# Inhomogeneous Superconductivity Onset in FeSe Studied by Transport Properties

**DOI:** 10.3390/ma16051840

**Published:** 2023-02-23

**Authors:** Pavel D. Grigoriev, Vladislav D. Kochev, Andrey P. Orlov, Aleksei V. Frolov, Alexander A. Sinchenko

**Affiliations:** 1L.D. Landau Institute for Theoretical Physics, 142432 Chernogolovka, Russia; 2Department of Theoretical Physics and Quantum Technologies, National University of Science and Technology ”MISiS”, 119049 Moscow, Russia; 3P.N. Lebedev Physical Institute of RAS, 119991 Moscow, Russia; 4Kotel’nikov Institute of Radioengineering and Electronics of RAS, 125009 Moscow, Russia; 5Institute of Nanotechnology of Microelectronics of RAS, 115487 Moscow, Russia; 6Laboratoire de Physique des Solides, Universite Paris-Saclay, 91405 Orsay, France

**Keywords:** FeSe, iron selenide, anisotropic superconductivity onset, heterogeneous superconductor, high-Tc, superconducting domain, twin boundary

## Abstract

Heterogeneous superconductivity onset is a common phenomenon in high-T${}_{c}$ superconductors of both the cuprate and iron-based families. It is manifested by a fairly wide transition from the metallic to zero-resistance states. Usually, in these strongly anisotropic materials, superconductivity (SC) first appears as isolated domains. This leads to anisotropic excess conductivity above T${}_{c}$, and the transport measurements provide valuable information about the SC domain structure deep within the sample. In bulk samples, this anisotropic SC onset gives an approximate average shape of SC grains, while in thin samples, it also indicates the average size of SC grains. In this work, both interlayer and intralayer resistivity were measured as a function of temperature in FeSe samples of various thicknesses. To measure the interlayer resistivity, FeSe mesa structures oriented across the layers were fabricated using FIB. As the sample thickness decreases, a significant increase in superconducting transition temperature T${}_{c}$ is observed: T${}_{c}$ raises from 8 K in bulk material to 12 K in microbridges of thickness ∼40 nm. We applied analytical and numerical calculations to analyze these and earlier data and find the aspect ratio and size of the SC domains in FeSe consistent with our resistivity and diamagnetic response measurements. We propose a simple and fairly accurate method for estimating the aspect ratio of SC domains from T${}_{c}$ anisotropy in samples of various small thicknesses. The relationship between nematic and superconducting domains in FeSe is discussed. We also generalize the analytical formulas for conductivity in heterogeneous anisotropic superconductors to the case of elongated SC domains of two perpendicular orientations with equal volume fractions, corresponding to the nematic domain structure in various Fe-based superconductors.

## 1. Introduction

The superconducting transition often occurs in a non-uniform way, when superconductivity (SC) initially appears in the form of isolated domains, which then acquire phase coherence leading to zero resistance. Such a heterogeneous SC onset attracts a great research activity and takes place in most high-temperature superconductors, including copper-oxide and iron-based families [1,2,3,4], where the spatial inhomogeneity of the SC energy gap has been directly observed in numerous scanning tunneling microscopy (STM) and scanning tunneling spectroscopy (STS) experiments [1,2,3,4,5,6,7,8,9,10,11,12,13,14]. The heterogeneous SC onset is also common to various organic superconductors [15,16,17,18,19,20,21], homogeneously disordered conventional superconductors [22], polycrystals [23], and many other materials.

The origin of such SC inhomogeneity in most cases is still debated. The possible reasons in crystalline compounds are the non-stoichiometry of chemical composition, uneven crystal growth, and the interplay of various types of electronic ordering, resulting in phase separation. The interplay with a spin- or charge-density waves probably leads to spatial inhomogeneity in organic metals [15,16,17,18,19] and in some cuprate high-T${}_{c}$ superconductors, e.g., in HgBa${}_{2}$CuO${}_{4+y}$ and La${}_{2}$CuO${}_{4+y}$ [4,24]. In all these materials, the density-wave heterogeneity was detected on a rather large length scale, ≳1 µm [4,17,24]. SC inhomogeneities of size ∼µm or more can be detected using scanning SQUID microscopy. For example, in La${}_{2-x}$Sr${}_{x}$CuO${}_{4}$ films 500 nm thick with $T_{c}=18$ K the diamagnetic domains of the size ∼ 5–200 µm were observed up to a temperature of 80 K $\gg T_{c}$ and were attributed to isolated superconducting islands as a precursor of SC onset [25].

As the temperature decreases, these superconducting islands become larger and, finally, occupy most of the area at $T\approx T_{c}$ [25]. Similar diamagnetic domains ∼100 µm in size were also observed in YBa${}_{2}$Cu${}_{3}$O${}_{6+x}$ films above $T_{c}$ [26].

In iron-based high-T${}_{c}$ superconductors, the spatial inhomogeneity of the SC energy gap is usually observed on a smaller length scale ∼10 nm using STM [3,7,8]. This inhomogeneity is probably due to non-stoichiometry and local variations in the chemical composition. However, in many Fe-based superconductors, including FeSe [11,27] and 122 iron pnictides *A*Fe${}_{2}$As${}_{2}$ (*A* = Ca,Sr,Ba) [9,10], there are also larger elongated domains [9,10,11] of width ≳30 nm and length ∼1 µm. They are related to the so-called ”nematic” phase transition from tetragonal crystal symmetry to orthorhombic one, driven by an electronic ordering [28,29,30]. In iron pnictides *A*Fe${}_{2}$As${}_{2}$, this transition is accompanied by antiferromagnetic ordering and occurs at $T_{SM}=173$ K for *A* = Ca, $T_{SM}=205$ K for *A* = Sr, and $T_{SM}=137$ K for *A* = Ba [10]. In FeSe, the compound studied below, this nematic transition happens at $T_{s}\approx 90$ K and is not related to any observed magnetic ordering [28,29]. Although its nature is still unknown, elastoresitivity measurements indicate that this nematic phase transition is an electronic instability [30].

The mechanism of spatial inhomogeneity, resulting from the nematic ordering, its microscopic structure, and its effect on SC properties are currently the subject of intensive research. Twin boundaries (TB) about 2 nm thick separate the nematic domains of orthogonal orientation both in 122 iron pnictides [31] and in FeSe [11]. In FeSe, SC is suppressed on TB [11], and magnetic vortices are pinned by TB, as visualized by STM [11]. The strain-induced detwinning increases the SC transition temperature $T_{c}$ by almost 1 K [30]. The stripes of orthogonal electronic ordering in FeSe were also detected by the so-called nano-ARPES [27], i.e. by ARPES with µm beam spot, but its energy resolution was insufficient to study superconducting properties. Point-contact spectroscopy revealed a significant local increase in T${}_{c}$ in FeSe [12], which may indicate that superconductivity in FeSe also onsets in the form of discrete domains. However, the typical size and shape of these domains were not determined in Ref. [12] because of the too-large area of point contact. The I–V curves, shown in Figure 1 of Ref. [12], indicate the local increase of $T_{c}$ up to 12–14 K, as compared to bulk $T_{c}\approx 8$ K, and strong SC fluctuations are observed up to 22 K.

The aforementioned experimental techniques—STM, STS, ARPES, and scanning SQUID microscopy—provide a direct visualization of SC heterogeneity, but share a common flaw in that they only measure the sample surface. However, the SC properties and domain structure may differ considerably on the surface of sample and deep in its bulk. Moreover, these experimental techniques provide no information about the domain size along the interlayer least-conducting *z*-direction, which is also important for understanding the electronic structure and superconducting properties of these materials. Hence, it would be very useful to study SC inhomogeneity deep inside the bulk sample. Although less visual, two such methods based on combined transport and diamagnetic-response measurements in bulk and finite-size samples were proposed recently [32,33,34,35,36,37], as described in the next section. The scenario of inhomogeneous SC onset in the form of isolated islands is supported by anisotropic excess conductivity in FeSe above $T_{c}$, measured in Refs. [32,33]. This anisotropy, in combination with the diamagnetic response data, provides information about the SC volume fraction and the averaged aspect ratio of SC domains. However, to analyze the experimental data on diamagnetic response, one needs to know the approximate size *d* of SC domains, at least if it is smaller than the penetration depth λ of the magnetic field into the superconductor. In the analysis of experimental data in Refs. [32,33], it was assumed that d≳λ, but this condition is violated if the SC domains are not larger than the nematic domains. The typical width of nematic domains is $d_{n}∼100$ nm, and their length exceeds λ, as observed by STM [14]. While the nematic domains in FeSe have been directly measured by STM/STS [11,13,14] and nano-ARPES [27], there is no any evidence that the SC and nematic domains in FeSe coincide. The size of SC domains in the conducting *x*-*y* plane can be measured by STS or point-contact spectroscopy, but as far as we know, no such experiments have been performed in FeSe, except for Ref. [12], where the spatial resolution was insufficient to study the SC domain shape and size.

The SC domain size along the *z* axis can be estimated from the SC percolation in finite-size samples, which corresponds to the onset of zero resistance in thin FeSe samples. A similar mechanism has recently been proposed to explain the anisotropic occurrence of zero resistance in organic superconductors [35]. On the other hand, the SC domain shape can be estimated from the anisotropy of excess conductivity above $T_{c}$ in combination with diamagnetic response data, similarly to Refs. [32,33,34]. Thus, the combination of these data helps to estimate the SC domain size in all directions and to answer the question of whether the SC and nematic domains coincide. This is important for understanding the mechanism of superconductivity in FeSe.

In Section 2, we describe the methods of both the experiment and the theoretical analysis. In Section 3, we generalize the Maxwell-Garnett approximation for elongated SC islands with two perpendicular orientations in anisotropic media. In Section 4, we present the results of our measurements of electronic transport in finite-size FeSe samples, used to estimate the interlayer size of SC domains, and the results of our numerical calculations of the percolation probability as a function of SC volume fraction ϕ and of sample thickness $L_{z}$ for the preliminary parameters of the domain aspect ratio. In Section 5, we desccribe a detailed theoretical analysis of the obtained and previous experimental data in order to extract useful information about the SC domain structure in FeSe. In particular, we numerically calculated the SC percolation threshold for the sample of relevant finite size and shape and compared it with our experimental data on resistivity to estimate the interlayer size of SC domains. In order to study the shape of SC domains, we also re-analyzed previously-obtained combined experimental data on excess conductivity and on diamagnetic response in FeSe above $T_{c}$ under the assumption that the SC domain width is d∼100 nm <λ. In Section 6, we present the main conclusions.

## 2. Materials and Methods

### 2.1. Experimental

Our experimental method was similar to that described in Ref. [32]. We used high quality platelike single crystals (flakes) of FeSe, grown in evacuated quartz ampoules at a permanent temperature gradient using AlCl${}_{3}$/KCl flux technique, as in Ref. [38]. The FeSe mesa structures (microbridges), as shown in Figure 1, were made using the focused ion beam (FIB) technology described in Refs. [39,40] from selected single-crystal samples of thickness of 2–4 µm (see Figure 1 and Figure 2a,b in Ref. [32]). Prior to FIB processing, a gold film was deposited by laser ablation to prepare the electrical contacts to the crystal. The electric resistance was measured in the conventional 4-probe configuration. In order to improve heat exchange, most of the structures were coated with collodium.

It is known that FIB may cause damage to the samples. The typical thickness of amorphous layer damaged by Ga ions in FeSe is about 50 nm. The minimum cross section of our mesa structures is 500×500 nm, the size presented in the article is 2×2 µm which is much larger than the expected depth of the damaged layer. We also evaluated the resistivity of all our structures and did not notice any strong change in the transport properties of thinner samples caused by FIB exposure. The obtained thin microbridges may crack during cooling. Such ”defect” samples are clearly visible under an FIB or an SEM [40]. Additionally, the damage to the sample during its cooling or measurement is easily detected by a sharp jump in resistance and by the analysis of its transport properties. Such ”defect” samples are excluded. Since FeSe mesas are rather fragile, we cooled them at a slow uniform rate of about 2 K/min. The cooling rate in the uncracked FeSe samples does not affect their transport properties.

### 2.2. The Origin of Anisotropic Resistivity Drop above $T_{c}$ and the Maxwell-Garnett Approximation

A new method, based on combined transport and diamagnetic-response measurements in bulk material, was recently suggested and applied to study the SC heterogeneity in several strongly anisotropic materials, including FeSe [32,33], YBa${}_{2}$Cu${}_{4}$O${}_{8}$ [34], and several organic superconductors [34,35,36]. These materials have a layered crystal structure and, hence, a strong anisotropy of electronic properties, which is typical of all ambient-pressure high-T${}_{c}$ superconductors. The resistivity drop above $T_{c}$ in all these compounds is anisotropic and strongest along the least conducting axis [16,17,18,19,32,33,41,42,43], which contradicts the standard theory [44,45] of superconducting fluctuations in homogeneous superconductors. This anisotropic effect of nascent superconductivity has been explained and analytically described [32,33,34] using a classical effective-medium model, namely, the well-known Maxwell-Garnett approximation (MGA) [46], generalized for strongly anisotropic heterogeneous metal with elliptical superconducting inclusions [32,33,34]. This simple model indeed predicts that an incipient superconductivity in the form of isolated domains in anisotropic conductors reduces the electrical resistivity anisotropically with a maximal effect along the least-conducting direction [32,33,34].

The qualitative picture behind this model [32,33,34] is very simple. In strongly anisotropic conductors with interlayer conductivity $\sigma_{zz}\ll \sigma_{yy}≲\sigma_{xx}$, the direct interlayer current perpendicular to the conducting layers is small, and the ratio is expressed by the parameter $\eta \equiv \sigma_{zz}/\sigma_{xx}\ll 1$. However, if SC emerges in a form of isolated domains, there is a second way of interlayer electric current via superconducting islands. If there are few of these domains, the major part of the current path goes in the normal phase. Instead of going along the weakly-conducting *z*-axis in the non-SC phase, this second path between the SC domains goes along the highly conducting layers, until it reaches another superconducting island, providing next lift across the layers (see Figure 2 for illustration). Then, there is no local current density along the poorly-conducting *z* direction in the non-SC phase. Hence, the contribution from this channel to interlayer conductivity does not contain the small anisotropy factor $\sigma_{zz}/\sigma_{xx}\ll 1$. Instead, this channel gives another small factor—the volume fraction ϕ of superconducting phase. The second way makes the main contribution to interlayer conductivity if ϕ/η≳1.

For the case of in-plane isotropy, $\sigma_{yy}=\sigma_{xx}$, and co-directional isolated spheroidal SC islands of volume fraction ϕ≪1, the analytical formulas for conductivity is rather simple:(1)$$\frac{\sigma_{xx}}{\sigma_{xx0}}\approx \frac{1}{1-\phi }+\phi ,\;\frac{\sigma_{zz}}{\sigma_{zz0}}\approx \frac{1}{1-\phi }+\frac{2\gamma^{2}\phi /\eta }{\ln\left(4\gamma^{2}/\eta \right)-2},$$
where $\gamma =a_{z}/a_{x}$ is the aspect ratio of main axes of spheroidal SC domains, and $\sigma_{xx0}$ is the in-plane conductivity in the absence of SC domains. Note that in Ref. [33] γ denoted the square of this aspect ratio. Equation (Equation 1) confirms the above qualitative picture: the interlayer conductivity $\sigma_{zz}$ indeed consists of two terms. The first term ${\left(1-\phi \right)}^{-1}$ coincides with that in $\sigma_{xx}$, while the second term contains a factor $\gamma^{2}\phi /\eta$ and at $\gamma^{2}/\eta >1$ determines the excess conductivity due to SC domains. Note that the domain size *d* does not enter Equation (Equation 1), which is valid for arbitrary distribution of *d*, provided their aspect ratio $a_{z}/a_{x}\equiv \gamma$ remains fixed. Equation (Equation 1) was generalized for fully anisotropic case $a_{z}\neq a_{y}\neq a_{x}$ and $\sigma_{xx0}\neq \sigma_{yy0}\neq \sigma_{zz0}$ in Ref. [34]. The averaged aspect ratios $a_{z}:a_{y}:a_{x}$ of SC islands may be extracted by comparing the measured conductivity with Equation (Equation 1), provided the SC volume fraction ϕ is known, for example, from diamagnetic response [32,33,34]. Alternatively, if the averaged aspect ratios $a_{z}:a_{y}:a_{x}$ of SC islands is known from the anisotropic diamagnetic response, the transport measurements can be used to extract the SC volume fraction ϕ as a function of some driving parameter, such as temperature, pressure, doping level, cooling rate, etc.

### 2.3. Numerical Calculations of Percolation Threshold

Another method [35], based on the transport measurements in finite-size sample, can be used to extract the averaged domain size. It was initially proposed to explain the anisotropic zero-resistance onset observed in organic superconductors, where zero resistance or a sharp resistance drop several times corresponds to a current percolation along the SC domains. Using this approach, we can calculate the SC volume fraction required for this current percolation for a given shape and size of the sample and of SC domains. However, for this method, the sample dimensions should be comparable or only several time greater than the SC domain dimensions. Otherwise, at the limit of an infinitely large sample, the percolation threshold will be isotropic, and we will not obtain information about the geometry of the domains.

The percolation probability p(ϕ) was calculated numerically using the Monte-Carlo algorithm. At each step, a random state with the proper number of spheroidal domains with a fixed size *d* and a fixed aspect ratio $\gamma =a_{z}/a_{x}$ was generated in a box of dimensions $L_{x}\times L_{y}\times L_{z}$, matching to our experiment. The required number of SC domains was determined by the volume fraction ϕ of the SC phase, and was selected in advance before the main simulation cycle. Each state corresponds to a graph whose vertices are SC domains. The vertices of the graph are connected by edges if the corresponding domains intersect. Thus, the problem of checking the presence of percolation along the axis is algorithmically reduced to finding the connected components of the graph containing vertices corresponding to the SC domains on the opposite edges of the sample, i.e. to the search for a percolation cluster. For each state, corresponding to one realization of SC islands, the percolation along each axis, i.e., the existence of a continuous path via intersecting SC domains, was checked, and the averaging over random realizations was made. About $10^{4}$–$10^{5}$ steps are usually sufficient to estimate the average percolation probability with an acceptable accuracy.

## 3. Generalization of Maxwell-Garnett Approximation

To consider elongated SC domains aligned in two perpendicular directions with equal probabilities, we have to generalize the MGA described above in Section 2.2. We start from the general equation for the effective conductivity tensor ${\tilde{\sigma }}^{e}$ of a heterogeneous media with M−1 types of unidirectionally aligned isotropic similar ellipsoidal inclusions inside an isotropic media with conductivity ${\tilde{\sigma }}^{1}$ (see Section 18 of Ref. [46]):(2)$$\sum_{j=1}^{M}\phi^{j}\left({\tilde{\sigma }}^{e}-{\tilde{\sigma }}^{j}\right){\tilde{R}}^{\left(j1\right)}=0,$$
where ${\tilde{\sigma }}^{j}=I\sigma^{j}$ is the effective conductivity tensor of the inclusions of type *j*, I is the 3 × 3 unity matrix, the so-called electric field concentration tensor:(3)$${\tilde{R}}^{\left(j1\right)}={\left[I+{\tilde{A}}^{\left(j\right)}\left(\sigma^{j}/\sigma^{1}-1\right)\right]}^{-1},$$
and ${\tilde{A}}^{\left(j\right)}$ is the diagonal depolarization tensor. For an ellipsoid with main semiaxes $a_{i}$ the depolarization tensor ${\tilde{A}}^{\left(j\right)}$ has only the diagonal components $A_{i}^{\left(j\right)}$ expressed via the integral (see Equation (17.25) of Ref. [46]):(4)$$A_{i}=\frac{a_{1}a_{2}a_{3}}{2}\int_{0}^{\infty }\frac{dt}{\left(t+a_{i}^{2}\right)\sqrt{\left(t+a_{1}^{2}\right)\left(t+a_{2}^{2}\right)\left(t+a_{3}^{2}\right)}},$$
where i=1,2,3 corresponds to x,y,z axes. The depolarization tensor has the property that its trace is unity, i.e., $\sum_{i}A_{i}=1$. The Equation (Equation 4) can be expressed via the elliptic integrals, as given by Equations (B1)–(B3) of Ref. [34].

In our case of elongated SC ellipses equally distributed along two perpendicular in-plane axes *x* and *y*, we have inclusions of two types j=2,3 with equal conductivity, $\sigma^{2}=\sigma^{3}=\infty$, and with equal volume fractions, $\phi^{2}=\phi^{3}=\phi /2$, but with different depolarization tensors ${\tilde{A}}^{\left(2\right)}\neq {\tilde{A}}^{\left(3\right)}$, because the elongated SC domains are differently aligned. Evidently, $A_{xy}^{\left(2\right)}=A_{yx}^{\left(3\right)}$. For SC domains with $\sigma^{2}=\sigma^{3}=\infty$, Equation (Equation 2) simplifies to:(5)$${\tilde{\sigma }}^{e}-{\tilde{\sigma }}^{1}=\frac{\sigma^{1}\phi }{2(1-\phi )}\left(\frac{1}{{\tilde{A}}^{\left(2\right)}}+\frac{1}{{\tilde{A}}^{\left(3\right)}}\right).$$

A similar result appears if one takes the SC ellipsoids randomly oriented in the *x*-*y* plane.

Due to the layered crystal structure, in the normal-metal phase FeSe is strongly anisotropic: the conductivity ratio $\eta \equiv \sigma_{zz}/\sigma_{xx}\approx 0.0025$. To describe such compounds with highly anisotropic conductivity ${\tilde{\sigma }}^{1}$ with diagonal components $\sigma_{ii}^{m}$, following the method used in Refs. [32,33,34], we applied the coordinate mapping:(6)$$x=x^{*},\;y=\sqrt{\mu }y^{*},\;z=\sqrt{\eta }z^{*},$$
where
(7)$$\mu =\sigma_{yy}^{m}/\sigma_{xx}^{m},\;\eta =\sigma_{zz}^{m}/\sigma_{xx}^{m},$$
with the simultaneous change of conductivity to ${\tilde{\sigma }}^{1}=\sigma^{m}I=\sigma_{xx}^{m}I$ in Equations (Equation 2) and (Equation 3). This mapping does not change the electrostatic continuity equation for the electric potential distribution inside matrix phase 1 of the heterogeneous medium:(8)$$-\nabla \cdot j=\sigma_{xx}^{m}\frac{\partial^{2}V}{\partial x^{2}}+\sigma_{yy}^{m}\frac{\partial^{2}V}{\partial y^{2}}+\sigma_{zz}^{m}\frac{\partial^{2}V}{\partial z^{2}}=0.$$

Hence, the voltage distribution in the original and mapped spaces are given by the same function: V(r). As a result of this mapping, the main semiaxes of SC inclusions change according to the rule:(9)$$a_{i}\to a_{i*}=a_{i}\sqrt{\sigma_{xx}^{m}/\sigma_{ii}^{m}},$$
and the tensors ${\tilde{R}}^{\left(j1\right)}$ and ${\tilde{A}}^{\left(j\right)}$ change to ${\tilde{R}}_{*}^{\left(j1\right)}$ and ${\tilde{A}}_{*}^{\left(j\right)}$ expressed by Equations (Equation 3) and (Equation 4) with the replacement in Equation (Equation 9). If initially the SC domains are not spherical but have ellipsoidal shape with the principal semiaxes $a=a_{1}$, $b=\beta a_{1}$ and $c=\gamma a_{1}$, then after the mapping to an isotropic media these domains keep an ellipsoidal shape but change the principal semiaxes to:(10)$$a_{1*}=a_{1},\;a_{2*}=a_{1}\beta /\sqrt{\mu },\;a_{3*}=a_{1}\gamma /\sqrt{\eta }.$$

In our case of FeSe μ=1, because $\sigma_{yy}^{m}=\sigma_{xx}^{m}$, and $1/\sqrt{\eta }\approx 20\gg 1$. Hence, after the mapping, we assumed $a_{z*}\gg a_{x*},a_{y*}$, but $a_{x*}\neq a_{y*}$ for elongated SC domains of the shape resembling that of nematic domains in FeSe. Then one may use the simplified formulas (B4)–(B6) of Ref. [34]:(11)$$A_{1*}\approx \frac{a_{2*}}{a_{1*}+a_{2*}}-\frac{a_{1*}a_{2*}}{2a_{3*}^{2}}\ln\frac{4a_{3*}/e}{a_{1*}+a_{2*}},$$
(12)$$A_{2*}\approx \frac{a_{1*}}{a_{1*}+a_{2*}}-\frac{a_{1*}a_{2*}}{2a_{3*}^{2}}\ln\frac{4a_{3*}/e}{a_{1*}+a_{2*}},$$
(13)$$A_{3*}\approx \frac{a_{1*}a_{2*}}{a_{3*}^{2}}\ln\frac{4a_{3*}/e}{a_{1*}+a_{2*}}.$$

Substituting Equations (Equation 11)–(Equation 13) for Equation (Equation 5), applying the mapping given by Equation (Equation 10) and using
(14)$$\frac{1}{A_{1*}^{\left(2\right)}}+\frac{1}{A_{2*}^{\left(3\right)}}\approx \frac{a_{1*}+a_{2*}}{a_{2*}}+\frac{a_{1*}+a_{2*}}{a_{1*}}=\frac{{\left(1+\beta \right)}^{2}}{\beta },$$
we obtain:(15)$$\frac{\Delta \sigma_{x}}{\sigma_{x}}=\frac{\Delta \sigma_{y}}{\sigma_{y}}\approx \frac{\phi }{\left(1-\phi \right)}\frac{{\left(1+\beta \right)}^{2}}{2\beta },$$
and
(16)$$\begin{array}{ccc} \frac{\Delta \sigma_{z}}{\sigma_{z}} & \approx & \frac{\phi }{1-\phi }{\left(\frac{a_{1*}a_{2*}}{a_{3*}^{2}}\ln\frac{4a_{3*}/e}{a_{1*}+a_{2*}}\right)}^{-1}\\ & = & \frac{\phi }{1-\phi }\frac{\gamma^{2}}{\eta \beta }{\left(\ln\frac{4}{e}\frac{\gamma /\sqrt{\eta }}{1+\beta }\right)}^{-1}, \end{array}$$
where e=2.71828. From Equations (Equation 15) and (Equation 16) we see that the relative excess conductivity is anisotropic,
(17)$$\frac{\Delta \sigma_{z}/\sigma_{z}}{\Delta \sigma_{x}/\sigma_{x}}\approx \frac{2\gamma^{2}}{\eta {\left(1+\beta \right)}^{2}}{\left(\ln\frac{4}{e}\frac{\gamma /\sqrt{\eta }}{1+\beta }\right)}^{-1},$$
which can be used to determine the aspect ratio $\gamma =a_{z}/a_{x}$ from transport measurement, provided another aspect ratio $\beta =a_{y}/a_{x}$ of SC domains is known.

Equations (Equation 15) and (Equation 16) are obtained for elongated SC islands, equally distributed along two main in-plane directions. This result is similar to the case of randomly oriented elongated SC islands, which is described by taking the trace of the matrix $R^{\left(j1\right)}$ [46]. Evidently, Equations (Equation 15)–(Equation 17) are invariant under the in-plane coordinate permutation x⟷y, which changes β→1/β and $\gamma \to \gamma /\beta^{2}$. Below we take $\beta =a_{y}/a_{x}\geq 1$, corresponding to SC domains oriented along *y*.

Let us compare Equations (Equation 15) and (Equation 16) with Equation (Equation 1) derived for spheroid SC islands at ϕ≪1, when the MGA is valid. For spheroid SC islands, when β=1, Equations (Equation 15)–(Equation 17) and Equation (Equation 1) give the same result: $\Delta \sigma_{x}/\sigma_{x}\approx 2\phi$, $\Delta \sigma_{z}/\sigma_{z}\approx$$\gamma^{2}\phi /\eta \ln\left(2\gamma /e\sqrt{\eta }\right)$. For β≠1 the relative excess conductivity $\Delta \sigma_{x}/\sigma_{x}$ in Equation (Equation 15) is greater than in Equation (Equation 1) at the same ϕ by a factor of ${\left(1+\beta \right)}^{2}/4\beta$, which considerably exceeds unity at β−1∼β. For β≫1 the increase $\Delta \sigma_{x}/\sigma_{x}\approx \phi \beta /2\propto \beta$, which has an evident physical interpretation: thin elongated SC inclusions of random orientation give the excess conductivity almost as if they were spheroid with the largest dimension $a_{y}=\beta a_{x}$, but the volume fraction ϕ in this case is smaller by a factor of $a_{x}/a_{y}=\beta^{-1}$. However, $\Delta \sigma_{z}/\sigma_{z}$ decreases at β≫1 by the factor $\beta^{-1}$. This is also evident: the increase of $a_{y}$ does not affect $\Delta \sigma_{z}$ but increases the SC volume fraction ϕ∝β. Hence, at the same ϕ, $\Delta \sigma_{z}\propto \beta^{-1}$.

## 4. Results

### 4.1. Experimental

The experimental results for the excess conductivity above $T_{c}$ and for diamagnetic response in bulk samples are given in Figures 2–4 of Ref. [32], and we do not show these data here. Nevertheless, we re-analyze these data below, taking the expected size of SC islands into account. Here we show the measured $R_{zz}\left(T\right)$ curves for thin samples, which may help to estimate the size of SC islands.

In Figure 2c of Ref. [32] the results for $R_{xx}\left(T\right)$ and $R_{zz}\left(T\right)$ measurements in the microbridge of thickness $L_{z}^{\left(0\right)}\approx 200$ nm is shown. One sees that the SC transition temperature $T_{c}$ itself is higher when determined from $R_{zz}\left(T\right)$ than from $R_{xx}\left(T\right)$. Similar $T_{c}$ anisotropy was reported in Ref. [37]. Below we explain this effect, analyze how this $T_{c}$ depends on sample thickness, and how this dependence can be used to extract information about the SC domains.

In Figure 3 we show the measured temperature dependence of normalized resistance $R_{zz}\left(T\right)/R_{zz}\left(T=15\;K\right)$ in several microbridge samples of the same in-plane size 2×2 µm${}^{2}$ but of different thickness, indicated in the figure legend for each curve. The normalization temperature T=15 K was chosen because (i) we expect negligible volume fraction and the corresponding effect of SC domains at T>15 K, and (ii) the $R_{zz}\left(T\right)/R_{zz}\left(T=15\;K\right)$ curves at T>15 K indeed coincide, as evidenced from Figure 3. The microbridge thickness for thicker samples is estimated visually from the SIM image of FeSe microbridge (overlap structure) oriented along the interlayer *c* axis, as shown in Figure 2b of Ref. [32] or in Figure 1a above. Therefore, we take the first microbridge thickness $L_{z}^{\left(1\right)}\approx 300$ nm with an error about 10%. The black curve in Figure 3 shows $R_{zz}\left(T\right)/R_{zz}\left(T=15\;K\right)$ in this microbridge. The SC transition temperature, corresponding to a 50% drop of resistance $R_{zz}\left(T\right)$, is about $T_{c}\left(50\%\right)\approx 8.5$ K for this sample, while a 90% drop of $R_{zz}\left(T\right)$ happens at $T_{c}\left(90\%\right)\approx 8$ K, which are only very slightly higher than $T_{c}$ determined from the in-plane resistance $R_{xx}\left(T\right)$ or from $R_{zz}\left(T\right)$ in bulk samples. This indicates that $L_{z}^{\left(1\right)}\approx 300$ nm $\gg d_{z}$, and this sample almost behaves as a bulk one for the SC onset.

The in-plane resistance $R_{xx}\left(T\right)/R_{xx}\left(T=15\;K\right)$ (not shown here) was measured only for larger samples, ∼1 μm thick, as in Figure 2a of Ref. [32]. It is quite close to that in Figure 2c of Ref. [32] and, more importantly, to the $R_{zz}\left(T\right)/R_{zz}\left(T=15\;K\right)$ curve in this microbridge of thickness $L_{z}^{\left(1\right)}\approx 300$ nm. If normalized resistivity along two axes is similar, $R_{xx}\left(T\right)/R_{xx}\left(T=15\;K\right)\approx R_{zz}\left(T\right)/R_{zz}\left(T=15\;K\right)$, from symmetry arguments one may conclude that the mean aspect ratio of SC domains $\gamma \equiv d_{z}/d_{x}\approx L_{z}/L_{x}$. This symmetry insight is confirmed by our percolation calculations for the spheroid SC domains below. For our FeSe sample it would give γ≈0.15.

For thinner samples in our experiment $T_{c}$ determined from $R_{zz}\left(T\right)$ is higher, while $T_{c}$ determined from $R_{xx}\left(T\right)$ in large samples almost does not change. For each of the thinner microbridges *m* the thickness $L_{z}^{\left(m\right)}$ was estimated according to:(18)$$L_{z}^{\left(m\right)}=L_{z}^{\left(1\right)}R_{zz}^{\left(m\right)}\left(T=15\;K\right)/R_{zz}^{\left(1\right)}\left(T=15\;K\right),$$
because for microbridges of the same in-plane area 2×2 µm${}^{2}$ the measured interlayer resistance is proportional to the microbridge thickness $L_{z}$. Unfortunately, this method of measuring microbridge thickness $L_{z}$ has an error, increasing with the decrease of $L_{z}$, because $L_{z}$ may slightly vary along the microbridge area 2×2 µm. This approach may underestimate $L_{z}$ by 10–20%, especially, for thinnest microbridges. Therefore, we take $L_{z}=50$ nm for our preliminary percolation calculations in the next subsection (see Figure 4a).

From Figure 3, we see that the SC transition temperature $T_{c}$ strongly increases, when the sample thickness $L_{z}$ decreases: from $T_{c}\approx 8$ K at $L_{z}^{\left(1\right)}\approx 300$ nm to $T_{c}\approx 12$ K at $L_{z}\approx 40$ nm. Note that $L_{z}\approx 40$ nm is still much larger than the in-plane SC coherence length $\xi_{0x}\approx 5\;\mathrm{nm}\gg \xi_{0z}$, so that the surface effects should not be important. We attribute this $T_{c}$ anisotropy to a heterogeneous SC onset and different percolation thresholds via SC domains in different directions for very thin samples, as described in the next subsection.

### 4.2. Preliminary Calculations of Anisotropic Percolation Probability

Figure 4a shows the calculated probability *p* of current percolation along the in-plane *x* and out-of-plane *z* axes via spheroid SC domains as a function of SC volume fraction ϕ for two different domain heights, $d_{z}=20$ nm and $d_{z}=5$ nm, in a sample of dimensions 2×2×0.2 µm${}^{3}$, close to our experiment. The aspect ratio $a_{z}/a_{x}=0.62$ of spheroid SC domains was chosen in agreement with Ref. [33]. Although we do not know exact domain shape and size, and the aspect ratio $a_{z}/a_{x}=0.62$ is corrected in the next section, we make several important conclusions from this calculation. First, (i) the percolation probability along the shortest sample dimension *z* is indeed much higher than along the other two directions, which explains the observed anisotropic SC transition temperature $T_{c}$ in thin FeSe microbridges. This result has a simple explanation: the percolation along the shortest sample dimension (thickness) requires a much smaller number of SC domains than along the longest dimension (length), as illustrated in Figure 4b. Second, (ii) the effect of $T_{c}$ anisotropy depends strongly on the SC domain size $d_{z}$ as compared to sample thickness $L_{z}$. The agreement with experiment is better for a larger domain size $d_{z}=20$ nm than for a smaller $d_{z}=5$ nm, suggesting an approximate average size of SC domains $d_{z}∼20$ nm. Third, (iii) the volume fraction $\phi_{c}$ of SC domains, required for current percolation in the thinnest sample, is still rather high: $\phi_{c}∼0.2$. From Figure 4a we find the sample-averaged percolation threshold $\phi_{c}$ as corresponding to percolation probability p=1/2.

## 5. Theoretical Analysis and Discussion

The SC volume fraction $\phi_{c}\approx 0.2$, corresponding the percolation threshold along *z* for thinnest sample in Figure 4, gives an estimate of the SC volume fraction at the SC transition temperature $T_{c}\approx 12$ K in this thinnest sample. The $\phi (T_{c}\approx 12\;K)\approx 0.2$ found in this way is much larger than the SC volume fraction $\phi \left(T=12\;K\right)<10^{-2}$ proposed in Refs. [32,33] basing on diamagnetic response data (see Figure 4d of Ref. [32]). A thinner sample, a larger size of the SC domains, or their elongated shape with a random orientation along *x* or *y* reduces $\phi_{c}$, but still keeps it large enough. Note that SC fluctuations can only enhance the diamagnetic response, further enhancing this discrepancy. This discrepancy is probably related to the assumption that the size $d_{x}$ of SC domains is larger than the SC penetration depth λ, which was made in the analysis of experimental data on diamagnetic response in Refs. [32,33]. In FeSe, the in-plane λ(T=0)≈400 nm and increases to ∼650 nm at $T\approx T_{c}=8$ K, as observed from $H_{c1}$ measurements (see Figure 6d of Ref. [47]). Therefore, if the width of SC domains does not exceed the width $d_{n}∼200$ nm of nematic domains, we have $d_{x}≲200\;\mathrm{nm}\ll \lambda$. If one assumes that the SC domains in FeSe are located inside the nematic domains and have a similar elongated shape of length $d_{y}>\lambda$, the diamagnetic response from SC islands can be estimated as the contribution of thin SC slabs ‖B of width $d_{x}≲200\;\mathrm{nm}\ll \lambda$ and volume fraction ϕ (see Equation (2.5) of Ref. [44]):(19)$$\Delta \chi \approx \left(\phi /4\pi \right)\left[\left(2\lambda /d\right)\tanh\left(d/2\lambda \right)-1\right].$$

At d≪λ, this simplifies to:(20)$$\Delta \chi \approx -\left(\phi /4\pi \right)\left(d^{2}/12\lambda^{2}\right).$$

If the SC domain length $d_{y}<\lambda$, one can estimate the diamagnetic response as a contribution from small SC spheres of volume fraction ϕ (see Equation (8.22) of Ref. [44] and Equation (17) of Ref. [34]):(21)$$\Delta \chi \approx \frac{\phi d^{2}}{10\pi \lambda^{2}\left(1-n\right)},$$
where the demagnetizing factor of SC islands n≪1 is small because of their oblate shape, and the factor (1−n) can be omitted. We see that Equations (Equation 20) and (Equation 21) produce similar results, differing only in a numerical coefficient ∼1.

The experimental data on diamagnetic response in Figure 4c of Ref. [32] give $\Delta \chi \left(T=12\;K\right)\approx 0.7\cdot 10^{-2}/4\pi$. The assumption d≫λ, implicitly made in Ref. [32], means that instead of Equations (Equation 19)–(Equation 21) the SC volume fraction ϕ and diamagnetic susceptibility Δχ are related by Δχ≈−ϕ/4π(1−n), which gives a strongly underestimated SC volume fraction $\phi_{1}\left(T\right)$, as shown in Figure 4d of Ref. [32]. In particular, it gives $\phi_{1}\left(T=12\;K\right)\approx 0.7\cdot 10^{-2}$, which is much smaller than the value ϕ(T=12K)≈0.2, expected from the SC percolation threshold along *z* axis for the thinnest sample in Figure 3 and Figure 4. The difference between ϕ(T) and $\phi_{1}\left(T\right)$ can be used to estimate the SC domain size. According to Equation (Equation 19), $\phi \left(T=12K\right)\approx 0.2$ and $4\pi \Delta \chi \left(T=12\;K\right)\approx 0.7\cdot 10^{-2}$ give an estimate $d_{x}/\lambda \approx 0.65$. Spheroid domain shape, according to Equation (Equation 21), gives a smaller domain diameter $d_{x}/\lambda \approx 0.3$, which better agrees with nematic domain width $d_{n}∼200$ nm. Note that, according to the BCS theory [44], λ(T) diverges at $T\to T_{c}$, but the $H_{c1}$ measurements give finite λ≈800 nm even at T=9 K [47]. The penetration depth λ(T) is, therefore, poorly defined for SC domains at $T>T_{c}$. If for our estimates we take λ≈800 nm, corresponding to $H_{c1}$ measurements at T=9 K [47], we than obtain $d_{x}∼0.3\lambda ∼240$ nm. Taking $\lambda =\lambda (T_{c})\approx 650$ nm [47] gives $d_{x}∼200$ nm. This SC domain size $d_{x}$ slightly exceeds the average nematic domain width $d_{n}∼100-200$ nm but is much less than the length of nematic domains. The inequality $d_{x}>d_{n}$ for averaged domain width is not too surprising. It can be explained by: (i) a significant fraction of wide SC and nematic domains with a width of $d_{n}≳200$ nm, which, due to their large size, make the main contribution to the diamagnetic response; (ii) the contribution of rare SC clusters consisting of several Josephson-coupled domains; (iii) additional diamagnetic response from SC fluctuations.

A weaker diamagnetic response Δχ(T) of small SC domains, given by Equations (Equation 19)–(Equation 21), corrects to a higher value the SC volume fraction ϕ(T) extracted from Δχ(T). It also corrects the estimated aspect ratio $\gamma =a_{z}/a_{x}$ of SC domains to a smaller value than $a_{z}/a_{x}\approx 0.62$ proposed in Ref. [33]. From Equation (Equation 16), neglecting the weakly dependent logarithmic factor, one obtains $\Delta \sigma_{z}/\sigma_{z}\propto \phi \gamma^{2}/\beta$. At a fixed measured excess conductivity $\Delta \sigma_{z}/\sigma_{z}$, this gives $\gamma \propto \sqrt{\beta /\phi }$. Hence, the ∼28 times increase of estimated ϕ, from $\phi \left(T=12\;K\right)\approx 0.7\cdot 10^{-2}$ to 0.2 decreases $\gamma \propto 1/\sqrt{\phi }$ about 5.3 times to $\gamma =a_{z}/a_{x}\approx 0.12$ as compared to the earlier proposed [33] value. The corresponding percolation calculations for the sample geometry as in our experiment and the SC domain size $d_{z}\approx 20$ nm with γ=0.12 are shown in Figure 5. These calculations suggest the percolation threshold $\phi_{c}\approx 0.12$ rather than $\phi_{c}\approx 0.2$ for the thinnest sample of $L_{z}\approx 40$ nm, where $T_{c}\approx 12$ K is determined from $R_{zz}\left(T\right)$ (see Figure 3). Hence, it corresponds to ϕ(T=12K)≈0.12 in FeSe. This slightly modifies the estimate of $\gamma \propto 1/\sqrt{\phi }$ to $\gamma \equiv a_{z}/a_{x}\approx 0.15$. The corresponding percolation calculations for γ≈0.15, $d_{z}\approx 20$ nm and for samples of different thicknesses, as in our experiment, are shown in Figure 6. The results of these calculations are in good agreement with the experimental data on anisotropic $T_{c}$ for FeSe microbridges, shown in Figure 3. In particular, in Figure 6, $T_{c}$ is almost isotropic for $L_{z}=L_{z}^{\left(1\right)}\approx 300$ nm, while for smaller $L_{z}$ $T_{c}$ is anisotropic and significantly higher if taken from $R_{zz}\left(T\right)$ rather than from $R_{xx}\left(T\right)$ curve, in agreement with Figure 3. Thus, our model of SC domain shape $\gamma =a_{z}/a_{x}\approx 0.15$ and size $d_{z}\approx 20$ nm now agrees with the available combined experimental data on anisotropic SC excess conductivity above $T_{c}$ in bulk samples [32,33], on the anisotropic SC transition temperature in thin FeSe microbridges (see Figure 3), and on diamagnetic response in FeSe above $T_{c}$ [32].

The current percolation and $T_{c}$ anisotropy in thin samples are very sensitive to the aspect ratio $\gamma =a_{z}/a_{x}$ of SC domains, which allows its accurate measurement. Indeed, for the aspect ratio $\gamma \equiv a_{z}/a_{x}=0.12$, as in Figure 5, the percolation threshold $\phi_{c}$ is isotropic for the sample thickness $L_{z}^{\left(2\right)}\approx 235$ nm, but not for $L_{z}^{\left(1\right)}\approx 300$ nm, as in Figure 6 for slightly different γ=0.15 and in our experiment. This suggests a new precise method for measuring the averaged aspect ratios $d_{x}:d_{y}:d_{z}$ of SC domains deep inside the sample, which is not accessible by STM or other surface measurements. Indeed, if small samples are fabricated, only several times larger than the expected domain size, then an anisotropic SC transition temperature $T_{c}$ to almost zero resistance should be observed, as in Figure 3. However, if the sample aspect ratios $L_{x}:L_{y}:L_{z}$ match the average aspect ratios $d_{x}:d_{y}:d_{z}$ of the SC domains, this $T_{c}$ anisotropy should disappear even if the sample size remains small, $L_{i}≲10d_{i}$, as for the sample with thickness $L_{z}=300$ nm and $L_{x}=L_{y}=2$ µm in our experiment. The aspect ratios of this sample give the aspect ratios of the SC domains.

Our experimental data in Figure 3 and percolation calculations in Figure 6 suggest the aspect ratio value $\gamma \equiv d_{z}/d_{x}\approx 0.15$ of SC domains in FeSe. This value agrees with the one obtained from the comparison of measured [32] excess conductivity and diamagnetic response at $T>T_{c}$ in bulk samples if the SC domain width $d_{x}∼200$ nm is comparable to the nematic domain width $d_{n}$ in FeSe. Note that the corresponding domain height $d_{z}=\gamma d_{x}∼30$ nm falls within the required interval 10 nm$<d_{z}<40$ nm, where, according to our percolation calculations, the $T_{c}$ anisotropy is significant for the sample thickness 40 nm$<L_{z}<300$ nm in our experiment. If the SC domains are not spheroid and have elongated shape with $\beta =a_{y}/a_{x}>1$, the estimated ratio $a_{z}/a_{x}$ grows $\propto \sqrt{\beta }$. For β=5 we obtain $\gamma =a_{z}/a_{x}\approx 0.25$. Percolation calculations can also be performed for this case. However, a direct experimental study of the shape and size of SC domains using STS measurements would be very useful in confirming our semiphenomenological predictions about the geometry of SC domains.

The methods employed above are also useful for many other compounds with heterogeneous superconductivity onset. For example, in FeS the spatial inhomogeneity has much larger length scale than in FeSe; the domain size d≈35 µm, far exceeding the SC penetration depth λ, was observed with submicrometer resolution spatially resolved ARPES (μ-ARPES) in FeS [48]. It is noteworthy that in FeS there is neither a ”nematic” phase transition to an orthorhombic lattice, which occurs in FeSe at $T_{n}\approx 90$ K and leads to a domain structure, nor magnetic ordering, as in FeTe below $T_{AFM}\approx 75$ K. Probably, the spatial inhomogeneity in FeS arises from the interplay of different types of electronic ordering, similar to organic superconductors.

## 6. Conclusions

In Figure 3, we present the experimental data on the temperature dependence of resistivity R(T) in thin FeSe samples, produced by cutting the bulk samples using FIB in the form of the microbridges shown in Figure 1. The SC transition temperature $T_{c}$ strongly increases as the sample thickness decreases. We explain this effect by calculating the percolation probability via the SC islands as a function of SC volume fraction ϕ in different directions. The anisotropy of the percolation threshold arises from the finite sample size and its flat shape. The thinnest microbridges are only a few times thicker than the SC domain size, and, in contrast to large samples, the percolation probability along the shortest sample dimension (thickness) is much higher than along its length (see Figure 4). Similar effects appear in organic superconductors [35]. Our calculations of percolation probability for the relevant sample shape and size, combined with our experimental data in Figure 3, suggest several important properties of the SC onset in FeSe. (i) The SC onset in FeSe is spatially heterogeneous and proceeds in the form of isolated SC domains, which become phase-coherent at lower temperature $T_{c}$, corresponding to SC transition of the entire sample. This is similar to many other high-Tc superconductors [1,2,3,4]; (ii) The SC domain height $d_{z}$ is several times smaller than the thinnest microbridge thickness $L_{z}\approx 40$ nm: $d_{z}∼20$ nm; (iii) The SC volume fraction at T=12 K in FeSe is rather large, ϕ∼0.1.

The small SC domain size $d_{z}$ means that the in-plane SC domain size $d_{x}∼7d_{z}$ is smaller than the penetration depth λ of magnetic field to FeSe superconductor. Hence, the estimates of the temperature dependence of the volume fraction $\phi_{1}\left(T\right)$, obtained from the diamagnetic response and shown in Figure 4d of Ref. [32] or in Figure 4 of Ref. [33], is underestimated by ∼20 times. As a result, the SC domain aspect ratio $a_{z}/a_{x}$ in Ref. [33] is, probably, overestimated ∼5 times. The combined new analysis suggests that the averaged aspect ratio $a_{z}/a_{x}\approx 0.15$, and the corresponding in-plane SC domain size $d_{x}\approx 100-200$ nm, which is comparable to the nematic domain width $d_{n}$ in FeSe. Thus, the hypothesis that SC domains are inside the nematic domains is consistent with the combined transport and diamagnetic experiments in bulk FeSe [32], as well as with our $R_{zz}\left(T\right)$ measurements in thin FeSe samples and corresponding percolation calculations. Notably, the proposed method of estimating the averaged SC domain aspect ratio $a_{z}/a_{x}$ from the anisotropy of SC transition temperature $T_{c}$ in finite-size samples turned out be very precise—with an accuracy ∼ 10%—and is applicable to many other heterogeneous superconductors.

We have also generalized the analytical formulas for conductivity in heterogeneous anisotropic superconductors in the case of elongated SC domains of two perpendicular orientations with equal volume fractions, corresponding to the nematic domain structure in various Fe-based superconductors (see Equations (Equation 15)–(Equation 17)). These formulae are useful for the analysis of anisotropic excess conductivity at $T>T_{c}$ in order to obtain useful information about the shape and volume fraction of SC domains.

In this paper, we focused on FeSe, although our method and discussion are applicable to other materials, including various cuprate and Fe-based high-T${}_{c}$ superconductors, organic metals, and other compounds. The obtained knowledge about the SC domains, electronic structure and SC properties of FeSe during the heterogeneous SC onset may help us to better understand the SC mechanisms and the properties of Fe-based superconductors, as well as to search for novel high-T${}_{c}$ superconductors.

## Figures and Tables

**Figure 1 materials-16-01840-f001:**
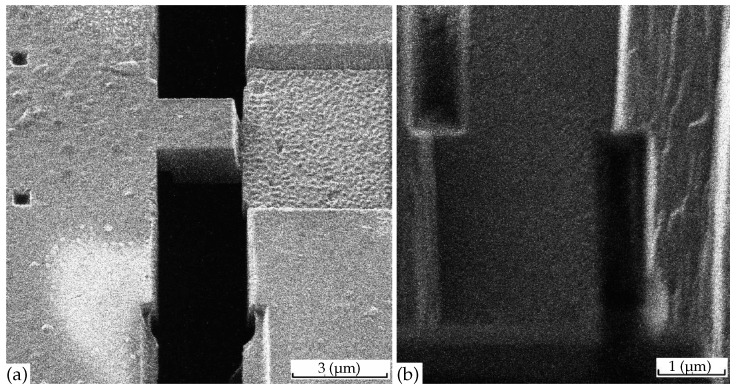
Photos of the microbridges, used in our experiment, from two different angles. (**a**) illustrates the 3D shape of our mesa structure, while (**b**) shows its dimensions.

**Figure 2 materials-16-01840-f002:**
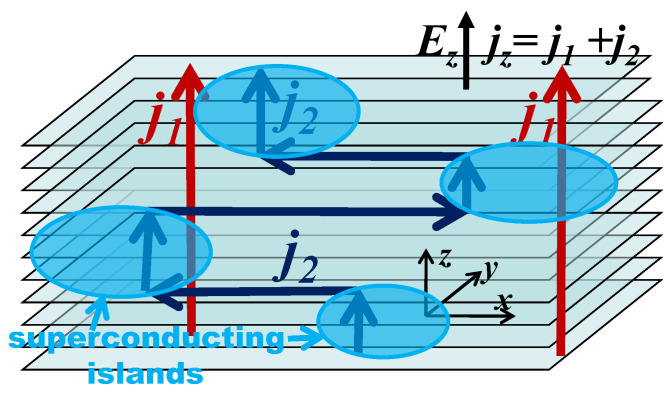
Illustration of two current channels, corresponding to two terms in Equation (Equation 1) for interlayer conductivity $\sigma_{zz}$ in a strongly anisotropic metal containing isolated superconducting islands.

**Figure 3 materials-16-01840-f003:**
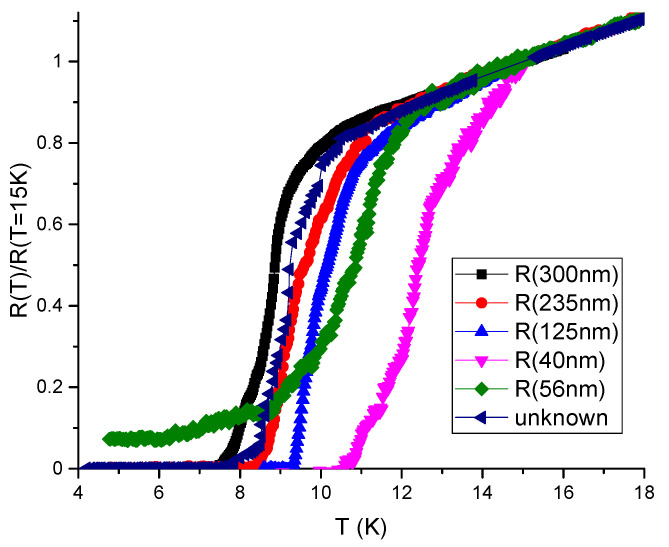
Measured temperature dependence of normalized resistance $R_{zz}\left(T\right)/R_{zz}\left(T=15K\right)$ in several samples of the same in-plane size 2×2 µm${}^{2}$ but of different thickness.

**Figure 4 materials-16-01840-f004:**
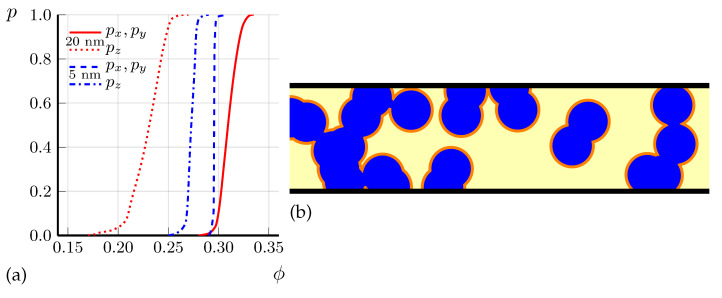
(**a**) Calculated probability *p* of current percolation along the in-plane *x* (solid and dashed curves) and the out-of-plane *z* axes (dotted and dash-dotted curves) via SC domains of spheroid shape with aspect ratio $a_{z}/a_{x}=0.62$ as a function of SC volume fraction ϕ for two different domain heights $d=d_{z}=20$ nm (red curves) and 5 nm (blue curves) in a sample of dimensions 2×2×0.2 µm${}^{3}$; (**b**) Two dimensional (2D) illustration showing that the current percolation along the sample thickness is easier than along the sample length. Circular SC islands (blue) with diameter *d* = 0.4 are randomly distributed inside a rectangular sample (yellow) of dimensions 7×2, forming SC channels between contact electrodes (black).

**Figure 5 materials-16-01840-f005:**
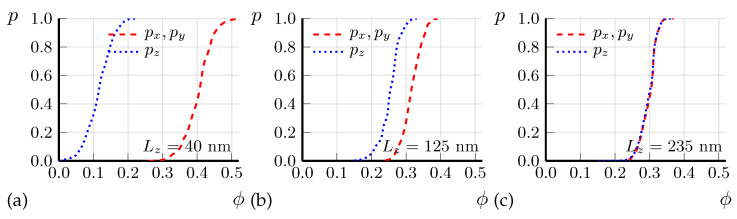
Calculated probability *p* of current percolation along the in-plane *x* (red dashed curves) and out-of-plane *z* (dotted blue curves) axes via the SC domains of spheroid shape with aspect ratio $a_{z}/a_{x}=0.12$ as a function of SC volume fraction ϕ in a sample of thickness $L_{z}=40$ nm (**a**), $L_{z}=125$ nm (**b**), and $L_{z}=235$ nm (**c**). The sample area in the conducting *x*-*y* plane is taken 2×2 µm${}^{2}$, as in our experiment. The domain height in our calculations is $d_{z}=20$ nm. The percolation probability is almost isotropic for the sample thickness $L_{z}=235$ nm at $a_{z}/a_{x}=0.12$.

**Figure 6 materials-16-01840-f006:**
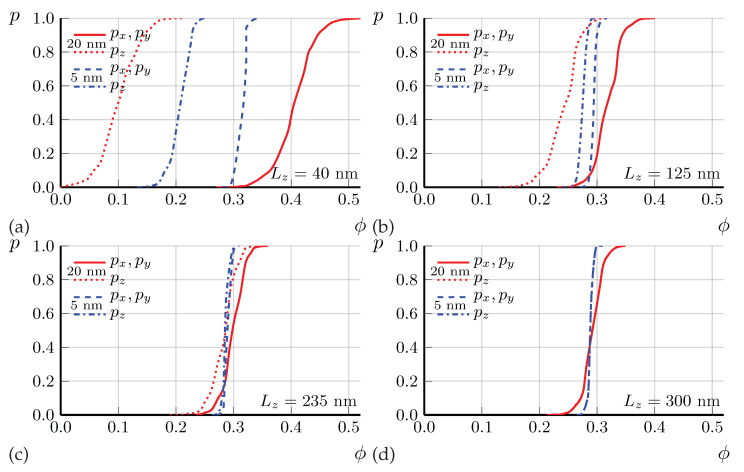
Probability *p* of current percolation along the in-plane *x* (red dashed curves) and the out-of-plane *z* axes (dotted blue curves) via SC domains of spheroid shape with aspect ratio $a_{z}/a_{x}=0.15$ and height $d_{z}=20$ nm as a function of SC volume fraction ϕ in a sample of *x*-*y* area 2×2 µm${}^{2}$, calculated for four different sample thicknesses $L_{z}=40$ nm (**a**), $L_{z}=125$ nm (**b**), $L_{z}=235$ nm (**c**), and $L_{z}=300$ nm (**d**). The percolation probability is isotropic for $L_{z}=300$ nm.

## Data Availability

Data are available from the authors.

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
