# Peer review of "Inhomogeneous Superconductivity Onset in FeSe Studied by Transport Properties"

_materials, 2023, doi:10.3390/ma16051840_

Round 1
Reviewer 1 Report
In the manuscript under review, the authors study the superconductivity of FeSe thin-films. Several FeSe samples, with varying thicknesses, were measured, and their resistances vs temperature are plotted in Fig. 3. In addition, the authors carry out theoretical modeling to understand the effects of anisotropy, film-thickness and inhomogeneity on the superconducting transition temperatures of FeSe thin-films. Although FeSe is an interesting superconductor and there is general interest in this compound, I cannot recommend the current manuscript for publication.
It is difficult to understand the manuscript. There are many instances in which readers have to guess what the authors intended to say. For example, in line 130, "A first method" which I could only interpret to mean "A new method". The quality of presentation has to be improved before the manuscript can be considered for publication.
Author Response
We thank the referee for the careful reading of our manuscript and sending us his/her comments. According to the referee suggestions, a major stylistic revision is made in the new version of our manuscript.
Reviewer 2 Report
This is an interesting work on an important subject appealing to a rather wide range of scientists. The manuscript is well written with detailed description, though it may have a room for further improvement of readability. I have just a few comments for the present authors' consideration in case of revision. (1) In the introduction, the authors refer to the organic superconductors with Refs. 15-19. However, all the refs. are old ones, as if the acitivity of the field was over. This reviewer recommend to add more recent studies on organic SCs with fluctuation and inhomogeniety such as Mater. Adv., 2022, 3, 1506. (2) The authors refer to "nematic" phase transition (L. 50) in the introduction. Although this manusctript is not a review, please consider to add a brief explanation on what the "nematic" phase is for general readers unfamiliar to this topic. It would add readability to this manuscript. (3) In Results and the succeeding sections, the units of "K" and "nm" are often written in italic without a space between values and them. (4) The authors discussed the obtained results in terms of percolation between SC domains, which appears to be reasonable in a way. However, the SC properties such as Tc's of the low-D SCs are sensitive to latitce defects including mechanical damage caused during the sample preparation and physical property measurements. They could produce artefact in the observed bahvior, which may be difficult to distinguish from the intrinsic behavior. Did not FIB cause such damage to the samples in this work? Did the authors confirm the crystalinity of the sample after the FIB by some experiment(s) to prove that they remain intact? Judging from the resistance data (Fig. 3), SC transitions are sharp with zero-resistance observed in some samples, while some appear to be not so sharp or did not exhibit zero-resistance to this reviewer. (5) Related to the comment (4), how about the effect of cooling rates on the observed behavior? How about the sample dependencies? Did all the samples with the same thickness reproducibly exhibit the same behavior within experimental error? (6) There are some mistypos. For example, please confirm L. 460.
Author Response
We thank the referee for the careful reading of our manuscript, for its positive evaluation and sending us useful comments. The manuscript is amended according to these comments.
(1) In the introduction two more recent studies on organic SCs with inhomogeneity are cited, including the paper Mater. Adv., 2022, 3, 1506, proposed by the referee.
(2) Although some explanation of the nematic phase transition was already given in the first version in the third paragraph, we extended this explanation and added two more references. In fact, the nature of the nematic ordering in FeSe is still unknown and debated.
(3) the units of "K" and "nm" are separated by a space from the numerical values in the entire manuscript.
(4) The referee is right that FIB may cause a damage to the samples. Typical thickness of amorphous layer damaged by Ga ions in such materials is about 50 nm. The minimum cross section of our mesa structures is 500x500 nm, the nominal size presented in the article is 2x2 µm which is much larger than the expected depth of the damaged layer. We also evaluated the resistivity of all our structures and did not notice a strong change in transport properties of thinner samples caused by FIB exposure.
The obtained thin microbridges may even have a crack. Such “defect” samples are clearly visible under a FIB or SEM (see IEEE paper). Also, damage to the sample during cooling or measurement is easy to detect by a sharp jump in resistance and analysis of the transport properties. The corresponding data obtained in “defect” samples are not shown in the manuscript. A broad superconducting transition is common for heterogeneous SC onset and does not necessarily mean a “defect” sample. Only the green curve in Fig. 3 may correspond to a slightly “defect” sample, because its resistance does not go to zero.
The corresponding explanation is added to the manuscript as a new last paragraph in Sec. 2.1.
(5) Related to the comment (4), how about the effect of cooling rates on the observed behavior? How about the sample dependencies? Did all the samples with the same thickness reproducibly exhibit the same behavior within experimental error?
Since FeSe mesas are rather fragile, we cooled them at a slow uniform rate of about 2K/min. The cooling rate in uncracked FeSe samples does not affect transport properties, in contrast to (TMTSF)_2ClO_4, for example.
The samples with the same thickness should not exhibit the same behavior within experimental error even in the ideal crystal, because the calculated percolation depends on the sample realization. Two different thin samples with the same thickness and the same SC volume fraction may have different resistivity and different Tc if the SC domains have different positions. Therefore, the calculated curves of percolation probability in Figs. 4-6 go from zero to unity in a wide interval of SC volume fraction. However, for large samples, as compared to SC domain size, where the averaging over domain realizations is more effective even within a single sample, this SC transition broadening decreases.
(6) Stylistic revision is made in the new version of our manuscript to improve its readability and to correct English and misprints.
Reviewer 3 Report
Report of the manuscript "Inhomogeneous superconductivity onset in FeSe studied by transport properties", by P.D. Grogoriev et al.
By discussing transport properties, the authors studied the onset of inhomogeneous superconductivity in FeSe. They observe an increase of the superconducting critical temperature Tc as the sample thickness decreases. By analyzing these data they discuss the size of the superconducting domains. The authors perform experiments and theory, and both methods are balanced. The topic is interesting as well as the results. I recommend the paper for publication.
Author Response
We thank the referee for the careful reading of our manuscript and for its positive evaluation. Stylistic revision is made in the new version of our manuscript to improve its readability. English is corrected.
Reviewer 4 Report
This paper intends to explain the observation of inhomogeneous superconducting onset in thin layered samples with the current applied along and perpendicular the to conducting layers.
The basic idea is based on a well established assumption that the percolation superconductivity is more probable to occur along the short sample dimension compared to the long one. The authors have developed the theory assuming the existence of elliptically shaped superconduction domains inside the nonsuperconducting matrix and have derived equations which could allow to relate the shape of the domains, the sample thickness and as well as the domain size to the onset of the superconducting transition in two perpendicular directions.
The theory was successfully applied to explain the thickens depedencies of the interlayer and intralayer superconductivity in FeSe samples.
I found the paper being well written and clearly understandable. I recommend the paper for publication as is.
Author Response

(The authors gave the same response as above.)

Reviewer 5 Report
The paper presents a large body of work that reports important data on size and shape of superconducting domains in FeSe. Their effect on the onset of bulk superconductivity and anisotropy of normal and superconducting properties is analyzed within a novel theoretical approach containing generalization of the Maxwell-Garnett approximation.
Fabrication of microbridges by a focused ion beam is a challenging experimental technique mastered by the authors. It is rather unique as is their well-argued theoretical explanation of the anisotropy of the temperature of the superconducting transition. Good theory can always be explained in simple terms. This has been masterfully done by the authors prior to doing Monte Carlo simulations of the percolation, by using simple expressions based upon the picture of superconducting domains.
The work is of exceptionally high quality. It is also of high importance for figuring out mechanisms behind phase diagrams of high-temperature superconductors. It reveals deep understanding of the physics of these materials by the authors and their comprehensive knowledge of literature.
The article is very well written. I recommend publication as is.
Author Response
We thank the referee for the careful reading of our manuscript and for its very positive evaluation. Stylistic revision is made in the new version of our manuscript to improve its readability. English is corrected.
Round 2
Reviewer 1 Report
The authors have revised the manuscript and improved its readability. Therefore, I recommend it for publication.